# geodl: An R package for geospatial deep learning semantic segmentation using torch and terra

**Aaron E. Maxwell**[1]*, **Sarah Farhadpour**[1], **Srinjoy Das**[2], **Yalin Yang**[1]

**1** Department of Geology and Geography, West Virginia University, Morgantown, WV, United States of America, **2** School of Mathematical and Data Sciences, West Virginia University, Morgantown, WV, United States of America

* Aaron.Maxwell@mail.wvu.edu

## Abstract

Convolutional neural network (CNN)-based deep learning (DL) methods have transformed the analysis of geospatial, Earth observation, and geophysical data due to their ability to model spatial context information at multiple scales. Such methods are especially applicable to pixel-level classification or semantic segmentation tasks. A variety of R packages have been developed for processing and analyzing geospatial data. However, there are currently no packages available for implementing geospatial DL in the R language and data science environment. This paper introduces the geodl R package, which supports pixel-level classification applied to a wide range of geospatial or Earth science data that can be represented as multidimensional arrays where each channel or band holds a predictor variable. geodl is built on the torch package, which supports the implementation of DL using the R and C++ languages without the need for installing a Python/PyTorch environment. This greatly simplifies the software environment needed to implement DL in R. Using geodl, geospatial raster-based data with varying numbers of bands, spatial resolutions, and coordinate reference systems are read and processed using the terra package, which makes use of C++ and allows for processing raster grids that are too large to fit into memory. Training loops are implemented with the luz package. The geodl package provides utility functions for creating raster masks or labels from vector-based geospatial data and image chips and associated masks from larger files and extents. It also defines a torch dataset subclass for geospatial data for use with torch dataloaders. UNet-based models are provided with a variety of optional ancillary modules or modifications. Common assessment metrics (i.e., overall accuracy, class-level recalls or producer's accuracies, class-level precisions or user's accuracies, and class-level F1-scores) are implemented along with a modified version of the unified focal loss framework, which allows for defining a variety of loss metrics using one consistent implementation and set of hyperparameters. Users can assess models using standard geospatial and remote sensing metrics and methods and use trained models to predict to large spatial extents. This paper introduces the geodl workflow, design philosophy, and goals for future development.

**Data Availability Statement:** The data and example code are available on FigShare: https://figshare.com/articles/dataset/geodl_example_data/23835165. A smaller version without the image chips used in the example training workflows is

also available on FigShare: https://figshare.com/articles/dataset/geodl_example_data_no_chips_/26824909. The geodl package has been published to the Comprehensive R Archive Network (CRAN) (https://cran.r-project.org/web/packages/geodl/index.html) and its source code is available on GitHub: https://github.com/maxwell-geospatial/geodl. It can be downloaded and installed using install.packages() or the remotes package. A website with associated articles and references for the package is available here: https://wvview.org/geodl/index.html.

**Funding:** Source 1: National Science Foundation (NSF) (Federal Award ID No. 2046059: "CAREER: Mapping Anthropocene Geomorphology with Deep Learning, Big Data Spatial Analytics, and LiDAR") Source 2: AmericaView and the U.S. Geological Survey under Grant/Cooperative Agreement No. G18AP00077.

# Introduction

## Need and justification

Pixel-level classification, also referred to as semantic segmentation within the computer vision community, has many applications in the geospatial sciences including land cover, forest type, and agricultural mapping. It is also used for differentiation of a class or feature of interest, such as buildings, ships, or landslides, from the surrounding landscape or background [1–3]. Such mapping or modeling tasks commonly rely on supervised learning, where predictor variables and pixel-level labels are used to train an algorithm to generate a model that can then be applied to new geographic extents to generate wall-to-wall predictions or maps. These methods employ geospatial data, structured as multidimensional arrays or data cubes. Representing geospatial data in a consistent multidimensional array format allows the use of a common supervised learning framework [4–6]. Data that can be represented in this format are collected from a variety of platforms (e.g., satellites, aircraft, or drones) and at a wide range of spatial and temporal resolutions. Further, a variety of data types can be represented as multidimensional arrays including true color, color infrared (CIR), multispectral, and hyperspectral imagery and synthetic aperture radar (SAR) backscatter. Additional predictor variables can be derived from individual images (e.g., band indices or principal components) or a timeseries of images (e.g., seasonal medians or coefficients generated from harmonic regression analysis). Other data sources include historic maps and other cartographic representations, land surface parameters (e.g., slope, topographic position index (TPI), topographic roughness index (TRI), and hillshades) derived from digital terrain models (DTMs), derivatives of light detection and ranging (lidar) point clouds (e.g., canopy height models (CHMs) and return intensity images), and subsurface geophysical measurements.

Semantic segmentation via supervised learning that relies purely on convolutional neural network (CNN)-based deep learning (DL) architectures were first introduced in 2014–2015 [7]. Such methods have been shown to be especially powerful due to their ability to use large amounts of labeled data in order to capture spatial context information at varying spatial scales and perform automatic feature extraction for classification tasks [8]. As a result, CNN-based methods are now replacing more established machine learning (ML) algorithms (e.g., support vector machines (SVMs), random forest (RF), and boosted decision trees (BDTs)) which have been traditionally used to label individual pixels or objects derived using geographic object-based image analysis (GEOBIA). CNN-based semantic segmentation has been operationalized and integrated into commercial geospatial software, including ArcGIS Pro [9] via the Image Analyst Extension [10] and ENVI [11] via the Deep Learning Module [12].

Open-source software, data science tools, and datasets have had a positive impact on how science is conducted, and the development of new tools and techniques has hastened the speed of scientific innovation and the transition of knowledge to action while also fostering reproducible and transparent research [13, 14]. Open-source software and datasets have reduced cost, increased the accessibility of research tools, and become key components of research and training infrastructure. Open-source DL tools (i.e., code to support data preparation; model creation, training, and validation; and inference to new data) are currently well developed in the Python [15] language and data science environment, resulting from the development of libraries including Tensorflow [16], Keras [17], and PyTorch [18]. This can be partially attributed to the development of DL amongst the computer vision research community, wherein Python is more popular than other common data science environments [19]. Base DL libraries, such as Tensorflow and PyTorch, have been extended with additional libraries specifically focused on semantic segmentation tasks. These libraries include Kornia [20], MMSegmentation [21], PixelLib [22], and Segmentation Models [23, 24]. The TorchGeo [25] and Raster

Vision [26] libraries support the use of geospatial data within the PyTorch ecosystem while eo-learn [27] supports processing Earth observation data with Python.

Many data, geospatial, and Earth scientists use R [28, 29] which was originally developed for statistical computing, data wrangling, and data analysis. This flexible environment offers a large number of specialized packages, familiarity and ease of use, quality of documentation, a large user base, and available integrated development environments (IDEs), such as RStudio [30]. Specialized R packages include those targeted for machine learning tasks such as clustering (e.g., dtwclust [31]) as well as those used for data wrangling (e.g., tidyverse [32]).

A large set of R packages have already been developed for reading, working with, and analyzing geospatial data specifically, such as sf [33], terra [34], stars [35], tmap [36], and sits, which supports Earth observation time series analysis [37]. The recent release of the terra package, which replaced the raster package [38] and currently (in early 2024) has had over six million downloads from the Comprehensive R Archive Network (CRAN) since its release in March 2020 based on download statistics obtained using the dlstats package [39], has improved computational efficiency for processing large raster grids, including digital elevation data and multispectral imagery [34]. Using the C++ language [40] via the rcpp package [41, 42], terra allows for reading in portions of large raster grids from disk as opposed to reading the entire dataset to memory, which has greatly improved the practical application of raster-based geospatial data handling and analysis in R [34].

Many DL tools in R rely on Python and act as a wrapper for Python libraries. Using the reticulate package [43], R packages such as keras [44], tensorflow [45], and fastai [46] allow for the execution of Python-based DL from the R environment using R code, and therefore requires the installation of Python environments and libraries. Highlighting the interest in implementing DL in R, the keras package has had over two million downloads from CRAN since its release in July 2017. The recently released torch package [47], which is written in R and C++ and built directly on libtorch (the PyTorch C++ backend) [48], simplifies the software stack by eliminating the need for the Python "middleman", thus avoiding reliance on Python and the associated issues stemming from incorrect versions of software or libraries and complications in setting up analytical environments, as well as the difficulties with troubleshooting errors. This is a large step forward in developing a DL experimentation environment and ecosystem native to R and C++. The torch package has been downloaded from CRAN over 160,000 times since its release in August 2020. We argue that there will be increased use of torch as an ecosystem of packages develops around it, as has occurred for the Python-based PyTorch implementation.

Currently, there are no R packages available specifically for DL applied to raster-based geospatial and Earth science data. This is problematic, as there are many demands and issues specific to geospatial and Earth science data, including the need to make use of raster data with varying numbers of channels or bands, maintain map coordinate reference information, assess models using discipline-specific methods and metrics, and merge results to generate wall-to-wall map products over large spatial extents. We argue that the torch and terra packages provide a unique framework to implement geospatial semantic segmentation in the R language and data science environment. The torch package provides an R/C++ implementation of DL that does not require Python/PyTorch while terra provides efficient handling and processing of large geospatial raster grids that may not fit into memory.

In this paper, we introduce the geodl package, which builds on torch and terra to support a general supervised learning, CNN-based semantic segmentation DL workflow that can be applied to a variety of geospatial data types structured in multidimensional arrays to characterize two-dimensional patterns to support pixel-level classification tasks. The package focuses specifically on semantic segmentation and is not designed to support scene classification,

object detection, or instance segmentation. It fills a key gap in the R environment and can ease the adoption of DL by geospatial scientists who have adopted R for research and applied mapping and modeling tasks. It provides utility functions to create raster masks from vector geospatial data, generate image chips from larger raster grids and associated masks, and collate image chip names and paths into R data frames; implements a generalized UNet-based framework with options to include a variety of ancillary modules; interfaces with the luz package to train models using loss metrics appropriate when class proportions are imbalanced and/or when difficult-to-predict samples should be prioritized; supports assessment of models using standard remote sensing methods and metrics; and allows for trained models to be applied to new data to generate map output.

The geodl package is available on CRAN (https://cran.r-project.org/web/packages/geodl/index.html) and can be installed in R using *install.packages()*. The source code is available on GitHub (https://github.com/maxwell-geospatial/geodl), and the package can alternatively be installed from GitHub using the remotes package [49]. Please consult the following page for details on how to install torch and its dependencies: https://torch.mlverse.org/docs/articles/installation. This page also links to documentation for configuring the environment for GPU-based computation, which is necessary to train CNN-based semantic segmentation models. The package's official website is available at https://wvview.org/geodl/index.html. The package webpage complements the material presented here by providing code and detailed vignettes demonstrating the package's workflow and associated functions. The data required to implement the examples have been provided via FigShare [50]: https://figshare.com/articles/dataset/geodl_example_data/23835165. A separate and smaller version of the data without the image chips used in the training workflow examples is also available on FigShare [51]: https://figshare.com/articles/dataset/geodl_example_data_no_chips_/26824909. More general torch examples are provided on the torch website: https://torch.mlverse.org/.

This article has two primary purposes. First, it provides an introduction to the geospatial semantic segmentation workflow implemented by geodl and introduces reference materials and documentation to help researchers and analysts implement the package for their own needs. Second, it provides a detailed documentation of the package's functions and design philosophy, including the configuration of the provided UNet architectures and their associated modules and the implementation and parameterization of the provided loss and assessment metrics. For specific code-based examples, please see the package webpage and associated vignettes.

## Data used in examples

In the examples provided on the package website and in this paper we make use of two datasets: topoDL [52] and landcover.ai [53]. We plan to add additional examples to the geodl webpage in the future to demonstrate more use cases and workflows using varying data sources. The topoDL dataset [52] was created by some of the authors of this paper and represents a binary semantic segmentation problem. Surface coal mining is denoted on topographic maps with a pink or brown pattern or symbology that is meant to represent surface disturbances. The topoDL dataset was developed to explore the use of semantic segmentation DL to extract the extents of historic surface mining from topographic maps [54]. The dataset consists of 123 1:24,000-scale, 7.5-minute topographic maps from the United States covering parts of eastern Kentucky, 23 covering parts of eastern Ohio, and 25 covering parts of southwestern Virginia. Mine extent masks were derived from the prospect- and mine-related features from U.S. Geological Survey 7.5- and 15-minute topographic quadrangle (version 10.0) dataset [55] generated by the USGS with some additional editing performed by the researchers. Only 7.5-minute

maps were used in the dataset. From the provided topographic maps and vector-based mine extent masks, it is possible to generate a large number of chips to train a DL model, as is demonstrated in the vignettes provided on the package webpage. These data are available on Fig-Share [52].

The Land Cover from Aerial Imagery, or landcover.ai, dataset [53] represents a multiclass classification problem in which five classes are differentiated: background, building, woodland, water, and road. Wall-to-wall pixel-level masks or labels were manually generated by the data originators using true color orthophotographs. Of the available photos, 33 have a spatial resolution of 25 cm, while eight have a resolution of 50 cm. A total area of 216.27 km$^2$ is mapped across different regions in Poland. These image extents can be divided into image chips and associated pixel-level masks using a Python script provided by the data originators. These data can be downloaded from the following website: https://landcover.ai.linuxpolska.com/.

## Overview of geodl workflow

In this section, we provide a general overview of the semantic segmentation workflow before discussing its functions and components in more detail in later sections. To aid readers in better understanding the general DL workflow, Table 1 explains some key terminology associated

**Table 1. Overview of deep learning terminology.**

| Term | Explanation |
|---|---|
| Chip | Image or raster-based predictor variable extent of a defined size (e.g., 64×64, 128×128, 256×256, or 512×512 cells) used as samples to train and validate models. |
| Mask or label | Raster-based reference data associated with each chip in which each cell is assigned an integer class label. |
| Training data | Data partition of chips and associated masks used to guide the parameter updates during the training process. |
| Validation data | Data partition of chips and associated masks which are commonly predicted at the end of each training epoch; used for hyperparameter tuning. |
| Test data | Withheld data partition of chips and associated masks used to assess the final model. |
| Loss metric | Measure used to guide parameter updates during the training process. The goal of the training process is to minimize this metric. Examples include cross entropy loss, Dice loss, and Tversky loss. |
| Assessment metric | Measure of model performance that can be monitored during the training process or calculated from the withheld test data. These metrics are not used to guide the parameter updates. Examples include overall accuracy, F1-score, recall, and precision. |
| Dataset | A torch class used to define how individual samples are processed before being provided to the dataloader and model. |
| Dataloader | A torch class that defines how mini-batches of data are provided to the model during training and inference. |
| Mini-batch | A subset of samples that are collectively provided to the model using a dataloader. Commonly, parameter updates are made after each mini-batch is processed as opposed to after one complete iteration over all training samples. |
| Epoch | One iteration over the entire training set or all mini-batches making up the entire training dataset. |
| Training loop | Process of iteratively training the model using the training data, a loss metric, backpropagation, and an optimization algorithm. |
| Optimizer | Algorithm used to update the model's trainable parameters. Examples include mini-batch stochastic gradient descent, RMSProp, and Adam. |
| Hyperparameter | A model or training process setting that is not learned during the training process. Examples include the optimizer's learning rate and the number of feature maps generated by each block in the encoder. |
| Parameter | A model component that is updated during the training process, such as weights and biases for convolutional kernels and gain and shift for batch normalization. |

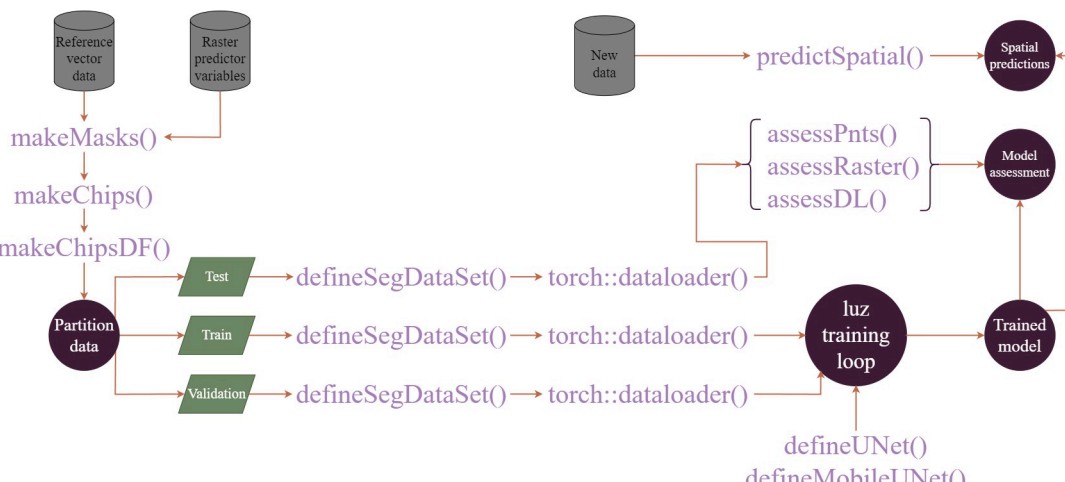

**Fig 1. Overview of geodl workflow.** Note that raster-based masks or labels can be used. The *makeMasks()* function is provided to convert vector-based reference data into raster masks.

with DL and the training and inference processes. Please see the package website (https://wvview.org/geodl/index.html) for examples of full workflows and applications of specific functions or routines.

CNN-based architectures attempt to capture spatial context information useful for differentiating classes by learning weights and biases associated with moving windows or kernels [8, 56, 57]. Since the goal is to learn local spatial context information, samples cannot be provided as individual cells. Instead, small image extents of a defined size, such as 64×64, 128×128, 256×256, or 512×512 cells, termed image chips, must be used. This is one of the key reasons that the CNN-based supervised learning workflow is different from the workflow implemented when using more traditional machine learning methods. The mask or label associated with each chip will consist of a single band or channel in which each cell is assigned an integer value representing one of the classes being differentiated.

Fig 1 conceptualizes the general workflow. It is common for labels to be provided as vector-based geospatial data as opposed to categorical raster data. The *makeMasks()* function allows for converting polygon vector geospatial data to raster grids. If raster-based masks are already available, this conversion is not necessary. Once raster-based predictor variables and raster-based labels are available, they can be broken into chips and associated masks of a user-defined size using the *makeChips()* function. This function generates a folder that contains the image chips and associated masks. The *makeChipsDF()* function generates an R data frame that lists all of the chips and associated masks within a directory.

Once a data frame is generated that lists all of the chips in a directory, they can be partitioned into non-overlapping training, validation, and test sets. It is also possible to generate image chips from different image extents and execute the *makeMasks()*, *makeChips()*, and *makeChipsDF()* functions separately for each partition as opposed to partitioning the data after the chips and data frame have been generated. The training set is used to guide the model parameter updates during the training process. The validation set is generally predicted at the end of each training epoch, or after one pass over the entire training set or once all training mini-batches are processed. The test set is reserved to assess the final model after the training process is completed.

In the torch environment, the dataset class is used to define how a single image chip and associated mask is processed before it is provided to the model while the dataloader class defines how the samples are aggregated into mini-batches to be passed to the model during the training loop [47]. The geodl package implements a dataset subclass, *defineSegDataSet()*, that allows for reading geospatial raster data using the terra package. The base *dataloader()* function from torch is used to define mini-batches since a unique subclass is not necessary for geospatial data.

The training process is implemented with the luz package [58], as is the standard within the torch R ecosystem. The use of luz simplifies the training loop. Once a trained model is generated, it can be assessed using the withheld test data and the *assessDL()* function, which accepts a trained model and a dataloader, generates assessment metrics, and aggregates the metrics across mini-batches. Assessment can also be conducted using point locations, *assessPnts()*, and raster-based predictions and associated raster-based reference data, *assessRaster()*. In order to generate raster predictions over larger spatial extents, a stack of raster-based predictor variables and a trained model can be provided to the *predictSpatial()* function.

## Details of package implementation

Now that we have provided a broad overview of the geodl semantic segmentation workflow, we will discuss the details of the implementation for those interested in understanding its inner workings and design philosophy. Fig 2 provides a detailed schematic of the geodl workflow.

Table 2 lists the package dependencies of geodl and their associated uses. As noted above, the torch package makes use of the libtorch C++ backend as opposed to PyTorch, so there is no need to install a Python environment, in contrast to other DL implementations in R. The terra package is used to read and generally handle raster geospatial data. Spatial reference information is maintained throughout the workflow, and raster grids with varying numbers of channels or bands can be efficiently read and processed. The luz package [58] simplifies the DL training loop, provides implementations of common callbacks (e.g., loggers, model checkpoints, learning rate modifiers, and early stopping), and allows for defining custom assessment metrics, a functionality that geodl makes use of to define new metrics, that can be monitored and aggregated over mini-batches during the training process. It also simplifies the placement and transfer of models and data between the central processing unit (CPU) and graphics processing unit (GPU). The torchvision package [59] provides additional functionality to complement torch for processing image data including applying image augmentations and implementing common computer vision architectures, such as the MobileNet-v2 architecture [60] used within geodl. The dplyr package [61], a key component of the tidyverse [32], is used for general data wrangling, manipulation, and summarization while sf [33] is used to read and process vector geospatial data. The MultiscaleDTM package [62] is used to define custom moving windows for calculating land surface parameters (LSPs) from digital terrain models (DTMs) within geodl's *makeTerrainDerivatives()* function while psych [63] is used for calculating summary statistics.

The DL workflow as implemented in open-source environments, such as PyTorch, is complicated by inconsistent data representations, dimensionality, and/or data types due to different developers using different conventions. For example, some loss functions require labels to be provided in a 32-bit float data type while others require a long integer data type. Single band, raster-based predictor variables can be represented as two- or three-dimensional arrays: [Width, Height] vs. [Channels/Predictors, Width, Height]. Similarly, associated labels can be stored with a [Class Indices, Width, Height] configuration or a [Width, Height] configuration. The geodl package adheres to the following standards:

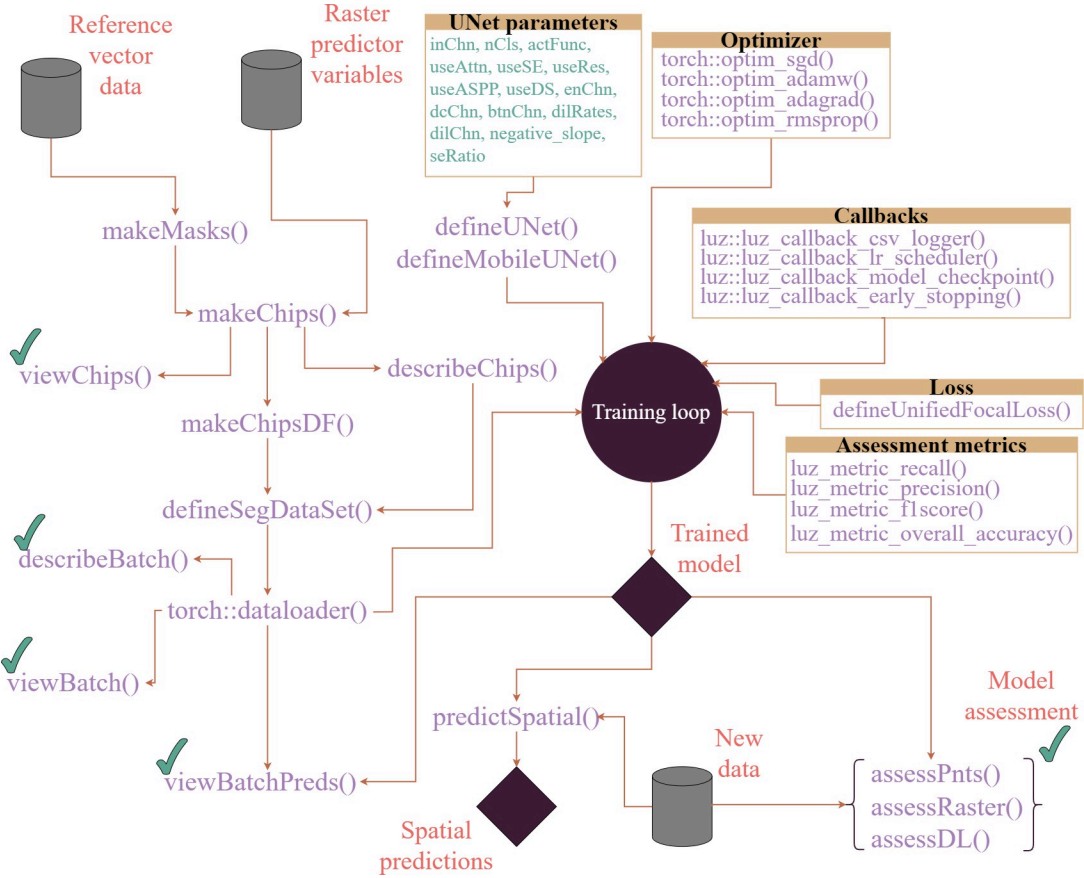

**Fig 2. Conceptualization of geodl DL semantic segmentation workflow.** Functions prefixed with *torch*:: are from the torch package while those prefixed with *luz*:: are from the luz package. All other functions are from geodl. Functions with a green check mark indicate those that are used as checks during the workflow or to assess trained models. Diamonds indicate outputs or results: the trained model and predictions to new raster data. Note that raster-based masks or labels can be used. The *makeMasks()* function is provided to convert vector-based reference data into raster masks.

1. All predictor variable tensors are expected to have a shape of [Channels/Predictors, Width, Height], even if only one predictor variable is provided, and to have a 32-bit float data type. All targets or labels are expected to have a shape of [Class Indices, Width, Height] and a

**Table 2. geodl package dependencies, uses, and associated references.**

| Dependency | Use | Reference |
|---|---|---|
| torch | Implement DL, tensor manipulations, computational graphs, neural network modules, and optimization algorithms | [47] |
| terra | Handle geospatial raster data | [34] |
| luz | Simplify training process, provide callbacks, implement assessment metrics, and handle transfer and placement of data on CPU and GPU | [58] |
| torchvision | Provide additional functionality for handling and processing image data with torch and apply data augmentations | [59] |
| dplyr | Generally wrangle, manipulate, and summarize data tables | [61] |
| sf | Process vector geospatial data | [33] |
| MultiscaleDTM | Generate moving windows of variable sizes and shapes for digital terrain analysis | [62] |
| psych | Calculate summary metrics | [63] |

long integer data type. Our *defineSegDataSet()* function processes input chips to meet these criteria and can accept a variety of raster-based input data; as a result, it is not difficult to adhere to the package standards.

2. All cases are treated as multiclass classification problems, even when only two classes are differentiated. This means that both positive and background logits are returned and logits are rescaled using a softmax as opposed to sigmoid activation for binary classification tasks. This design decision was made to standardize and simplify the implementation of losses and assessment metrics. Class weightings can be used to specify the relative weightings of classes in loss and assessment metrics. When a positive class is being differentiated from the image background, weightings can be useful for controlling the relative importance of predicting the positive and background classes.

3. Class indices can start at zero or one. However, no integer values can be skipped. Since R begins indexing at one and due to the use of one-hot encoding in some components of the package, zero index values can cause errors. As a result, sometimes it is necessary to adjust indices such that they start at one. This is the purpose of the *zeroStart* parameter used in many of the implemented functions.

## Data preparation and utilities

It is common for reference labels to be generated as geospatial vector data and stored within a geospatial vector data format, such as a feature class within a file geodatabase, a shapefile, or a layer within a GeoPackage. As a result, it is necessary to provide utilities to convert vector data into categorical or integer raster grids where unique indices differentiate each class or the class of interest and the background. The *makeMasks()* function serves this purpose within geodl; it generates raster masks that align with the available raster predictor variables (i.e., have the same coordinate reference system, spatial resolution, origin, extent, and number of rows and columns of cells). It can also crop predictor variable raster grids and generated masks to a defined extent, as defined by a vector-based polygon boundary. A column in the vector layer attribute table is used to define the class codes. For binary classification problems, the background class should be coded as zero and the presence or positive class should be coded as one. If masks are already represented as raster grids, this conversion process is not required.

The *makeChips()* and *makeChipsMultiClass()* functions are used to generate chips and associated masks. The *makeChips()* function is used for two-class problems where the positive class is assigned a value of one and the background class is assigned a value of zero. When more than two classes are differentiated using unique numeric codes, the *makeChipsMultiClass()* function should be used. If the data are sparsely labeled (i.e., not all pixels have class labels even though they belong to a specific class), these pixels should be assigned a unique numeric code that can then be flagged in the loss and/or assessment metric(s) to be assigned a weight of zero, and thus ignored. When using *makeChips()*, all chips can be generated, just those containing at least one pixel mapped to the positive case, or both background-only and positive case chips, which are written to separate folders. This allows the user to control whether all background-only chips are used in the training and/or validation process, or whether only a subset of background-only chips are used.

Once chips and associated masks are written to disk, the *makeChipsDF()* function is used to list the names of each chip and associated mask into an R data frame and, optionally, a comma-separated values (CSV) file written to disk. If positive and background-only chips are differentiated, a column is added to the data frame to denote this, which the user can use to filter or subsample the available chips. The *viewChips()* function plots a random set of chips and

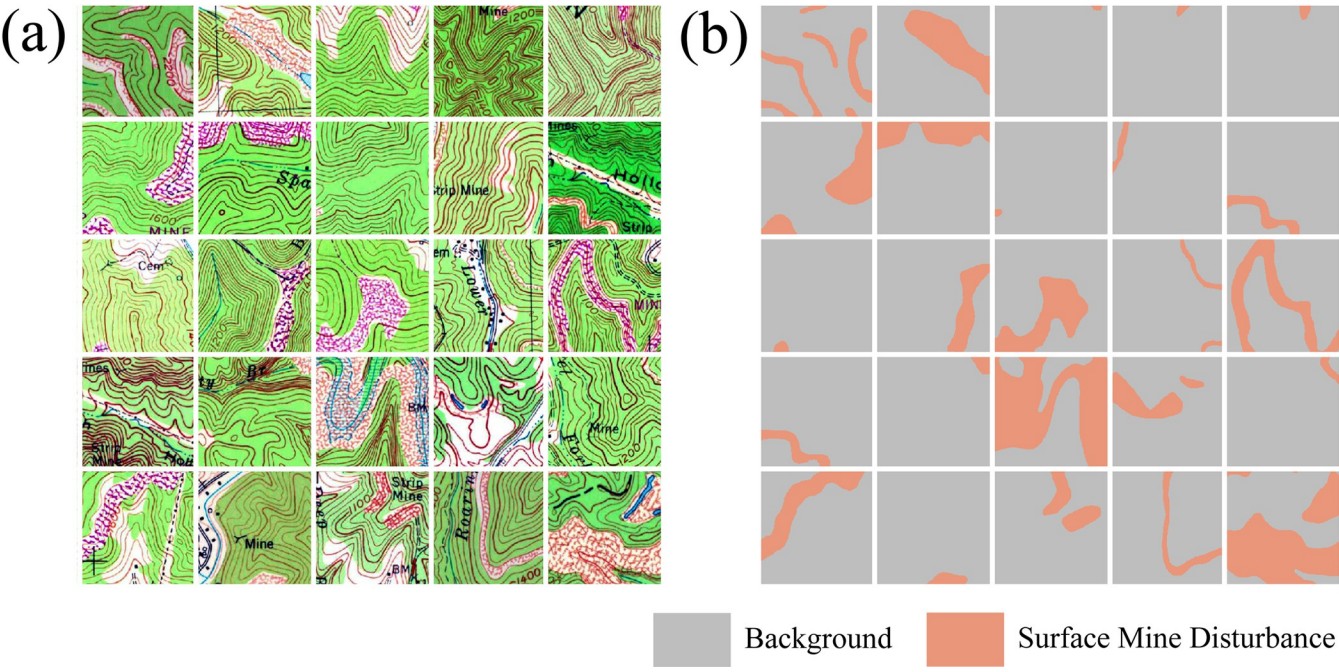

**Fig 3. Output from *viewChips()* function for surface mine disturbance extraction from topographic maps using topoDL dataset [52].** (a) image chips; (b) reference masks.

associated masks from the specified directory. Figs 3 and 4 provide example outputs from this function for the topoDL and landcover.ai datasets, respectively. In order to apply normalization and/or estimate the relative proportion of classes within the dataset, which can be useful

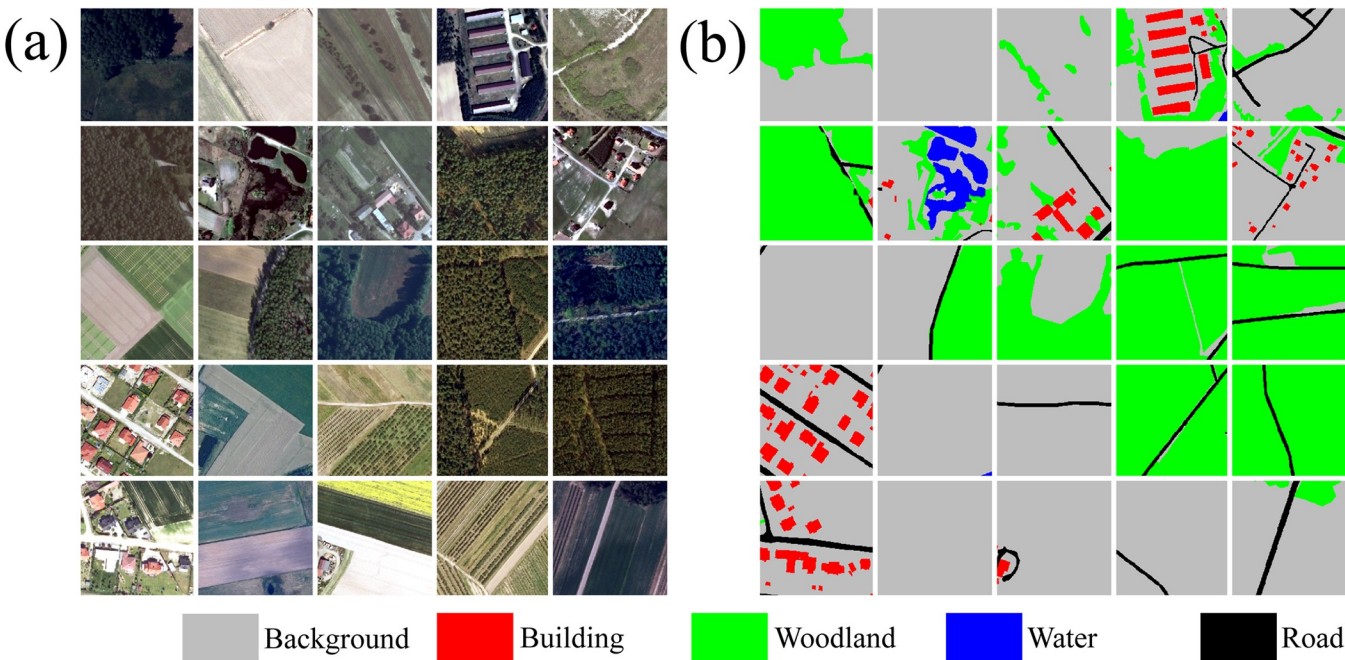

**Fig 4. Output from *viewChips()* function for general land cover mapping from landcover.ai dataset [53].** (a) image chips; (b) reference masks.

for applying class weightings within loss metrics, statistics must be calculated from the chips and associated masks. This is the purpose of the *describeChips()* function.

## Datasets and data augmentations

The geodl *defineSegDataSet()* function is implemented by subclassing the dataset class defined within torch. It accepts a data frame created by the *makeChipsDF()* function to read chip and mask files from disk and generate tensors. It also provides a set of options for performing data rescaling (by dividing by a specified value), normalization to *z*-scores using band means and standard deviations, and random augmentations. Random augmentations are implemented with torchvision and include horizontal or vertical flips and augmentations of brightness, contrast, gamma, hue, and saturation. The user is able to specify the probability that an augmentation will be performed, the maximum number of augmentations to apply to a single chip, and the range of augmentation-specific parameters from which to select a random value. The goal of performing these augmentations is to potentially combat overfitting [64, 65]. Note that if a chip is flipped, the mask will also be flipped to maintain alignment. Also, some transformations are not possible for all data types; for example, changes in hue and saturation are only applicable to RGB data.

Once an instance of *defineSegDataSet()* is instantiated, it can be provided to the *dataloader ()* function from torch to define a dataloader. A mini-batch of predictor variables and associated masks provided by the dataloader can be visualized using *viewBatch()* while *describeBatch ()* provides a check of a mini-batch by returning the mini-batch size; data type, dimensionality, and shape of the predictor variables and masks tensors; predictor variable means and standard deviations; and count of pixels mapped to each class index.

## UNet-based models

**Model overview.**   This section describes the UNet implementations provided by geodl. The UNet architecture was proposed in 2015 by Ronnenberger et al. [66] for semantic segmentation of biomedical imagery. Since its inception, it has expanded into a more general framework. UNet-like architectures share several common components; they consist of an encoder that is used to learn spatial patterns or context at multiple scales via learnable convolution kernels that are applied to input data or prior feature maps to generate new feature maps. The encoder is broken into separate blocks that consist of 2D convolution layers, activation functions (e.g., rectified linear unit (ReLU)), and, commonly, batch normalization. Each block is separated by a max pooling operation, which reduces the size of the array in the spatial dimensions and aids in allowing for learning patterns at varying spatial scales. The bottleneck separates the encoder and decoder components and represents the stage at which the data have been reduced to the smallest spatial resolution within the architecture.

The purpose of the decoder is to restore the spatial resolution of the data in order to make pixel-level predictions as opposed to scene-level predictions. Similar to the encoder, the decoder is separated into blocks consisting of 2D convolution layers, activation functions, and batch normalization. Instead of decreasing the size of the array in the spatial dimensions using max pooling, the array is upsampled using either resampling algorithms, such as bilinear interpolation, or transpose convolution. Between encoder and decoder blocks with the same spatial resolution, skip connections are added that allow for semantic information to be shared across the model. Lastly, the pixel-level classification is performed using 1×1 2D convolution to return logits for each differentiated class [66].

The package's *defineUNet()* function provides a flexible means to generate a UNet-like architecture for semantic segmentation tasks. This architecture is conceptualized in Fig 5 and

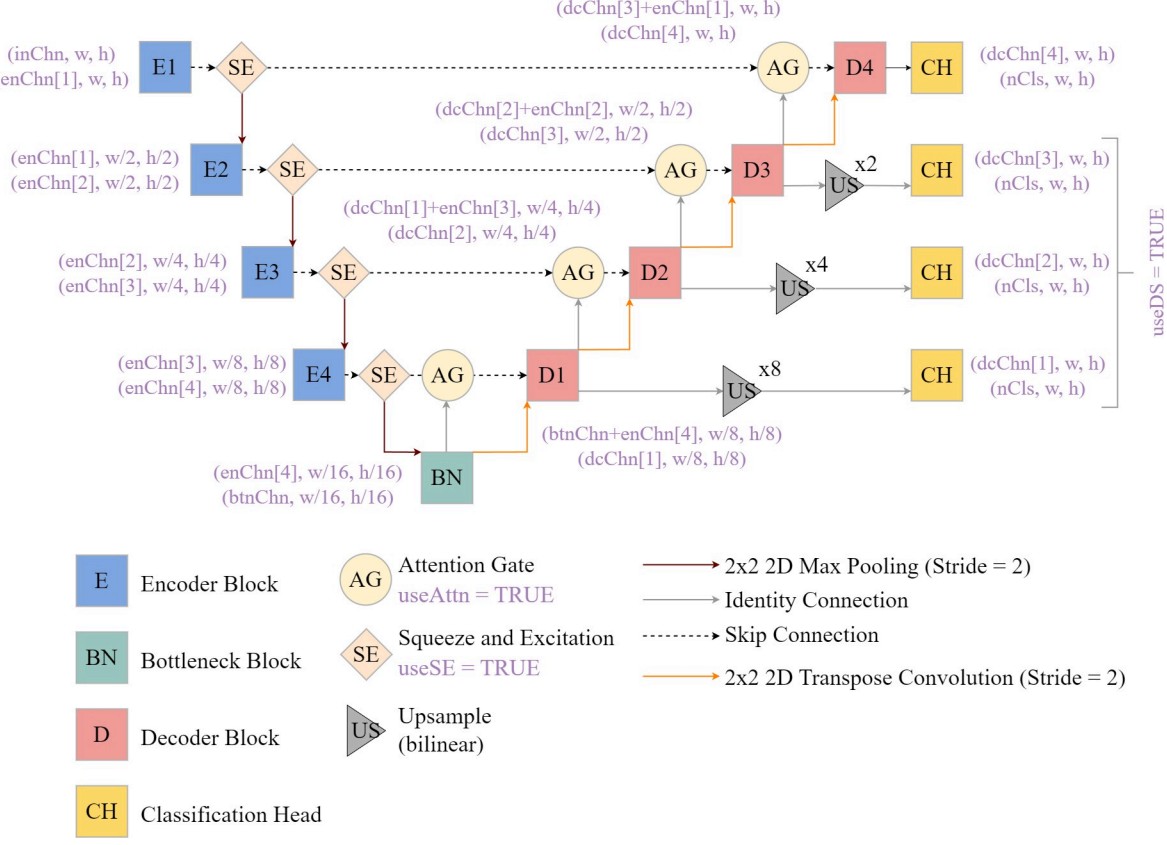

**Fig 5. UNet architecture implemented in geodl and associated modules.** Blocks prefixed with E represent encoder blocks while those prefixed with D represent decoder blocks. w = width, h = height. Purple text corresponds to function parameters as implemented in geodl. See Table 3 for explanations of function parameter names.

its associated parameters are described in Table 3. It can accept a variable number of input predictor variables (i.e., does not require RGB or three-band input data) and output classes. It contains four **encoder** blocks, a bottleneck block, and four decoder blocks, and the user can specify the number of output feature maps from each block. By default, rectified linear unit (ReLU) activation functions are implemented throughout the architecture to incorporate non-linearity, and batch normalization is used to combat gradient issues and aid in convergence. In the encoder, the size of the array in the spatial dimensions is reduced by half following each block using 2×2 max pooling with a stride of two. In the decoder, 2D transpose convolution is used to double the spatial resolution of the feature maps provided from the prior block, also using a stride of two. The final class logits are predicted using 1×1 2D convolution.

A variety of optional configurations or modules can be added to the architecture including residual connections, squeeze and excitation modules, attention gates, a modified atrous spatial pyramid pooling (ASPP) module as the bottleneck block, and/or deep supervision. These additional modules are described in the following sections.

**Activation functions.** The ReLU activation function is used by default within the architecture. This function simply converts all negative activations to zero and maintains all positive activations as their original value (Eq 1) [8, 67]. To combat the "dying ReLU" problem, it may be desirable to maintain negative activations, but with a reduced magnitude. Leaky ReLU accomplishes this by maintaining positive activations and multiplying negative activations by a positive value smaller than one (called a negative slope term (nst) in Eq 2) in order to reduce

**Table 3.** *defineUNet()* function parameters and associated explanations.

| Parameter | Explanation |
|---|---|
| *inChn* | Number of input channels or predictor variables |
| *nCls* | Number of classes being differentiated |
| *actFunc* | Activation function to use (ReLU, leaky ReLU, or swish) |
| *useAttn* | Whether or not to include attention gates along skip connections |
| *useSE* | Whether to include squeeze and excitation modules in the encoder blocks |
| *useRes* | Whether or not to include residual connections throughout the architecture |
| *useASPP* | Whether or not to replace the bottleneck block with an ASPP module |
| *useDS* | Whether or not to use deep supervision |
| *enChn* | Number of output feature maps produced by each encoder block |
| *dcChn* | Number of output feature maps produced by each decoder block |
| *btnChn* | Number of output feature maps produced by bottleneck block |
| *dilRates* | Dilation rates to use in ASPP module |
| *dilChn* | Number of feature maps produced by each branch in the ASPP module |
| *negative_slope* | Negative slope term to apply if leaky ReLU is used |
| *seRatio* | Squeeze and excitation reduction ratio |

their magnitude (Eq 2) [68]. Another option is the swish activation, which is calculated by multiplying the activation by the activation modified using a sigmoid function (Eq 3) [69].

$$ReLU = \begin{cases} activation \; if \; activation > 0 \\ 0 \; if \; activation \leq 0 \end{cases} \tag{Eq1}$$

$$Leaky \; ReLU = \begin{cases} activation \; if \; activation > 0 \\ activation*nst \; if \; activation \leq 0 \end{cases} \tag{Eq2}$$

$$Swish = activation \cdot \mathrm{sigmoid}(activation) \tag{Eq3}$$

**Residual connections.** The traditional double-convolution layers used in UNet consist of passing the input predictor variables or feature maps produced from prior layers through a 3×3 2D convolution block to produce a set of feature maps equal to the number of input feature maps. These results are then passed through a second 3×3 2D convolution layer to generate the user-defined number of output feature maps for that stage in the architecture [66]. This is conceptualized in Fig 6(A).

A residual connection or residual block augments this architecture by adding the input feature maps directly to the output from the second 2D convolution layer (Fig 6(B)). The goal is to potentially reduce the vanishing gradient issue by maintaining this original signal provided to the block in the output of the block [70]. Note that the input data are augmented along the residual path so that the number of input feature maps matches the number of feature maps generated by the convolution operations, since this is required to add the input and output tensors.

**Squeeze and excitation module.** The goal of a squeeze and excitation (SE) module is to capture interrelationships between channels or feature maps [71]. Fig 7 provides a conceptualization of this module. Fig 7(A) shows a version of the module that does not include a residual connection while Fig 7(B) does include a residual connection. First, each input channel is reduced to a single value using global average pooling, resulting in a vector of input channel or

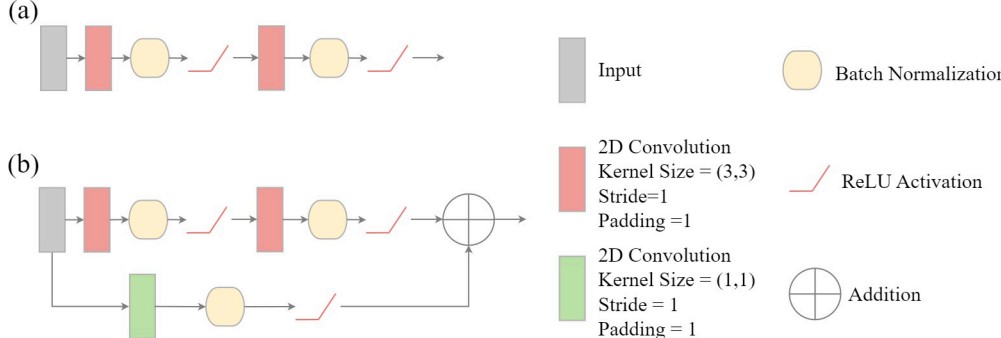

**Fig 6.** (a) double convolution block. (b) double convolution block with residual connection. Double convolution blocks are the primary components of the encoder and decoder blocks within the UNet architecture.

feature map means. This is the "squeeze" component of the module. Following the global average pooling, the remainder of the module is the "excitation" component where the rescaling of the input data is guided by the learned interrelationships between the channels. First, the means are modified using a fully connected layer, ReLU activation, and a final fully connected layer. The goal of this sequence of operations is to model non-linear interrelationships between the means. The output from the last fully connected layer is then passed through a sigmoid activation function to rescale the values to a range of zero to one. The input channels or feature maps are then multiplied by the rescaled values on a per channel basis to augment the input data.

**Attention gates.** An attention gate (AG) module provides a mechanism to allow for forcing the model to focus on key features or regions within the image [72, 73]. The idea is to use the results from the subsequent layer in the network, where a deeper set of features have been extracted, to add focus, or attention, to the feature maps from the prior layer that are then concatenated with the upscaled feature maps from the following block and fed to the decoder block. This process is conceptualized in Fig 8. The feature maps from the next layer in the sequence (for example, the feature maps produced by decoder block three when the attention gate is applied to the feature maps from encoder block one) are passed through a 1×1 2D convolution layer with a stride of one and a padding of zero, and the number of channels is changed to match those from the prior block. A batch normalization is then applied. The feature maps from the current layer are passed through a 1×1 2D convolution layer with a stride of two and a padding of zero, and the number of output feature maps is equal to the number of input feature maps. Since a stride of two is used, the spatial resolution is reduced by half such

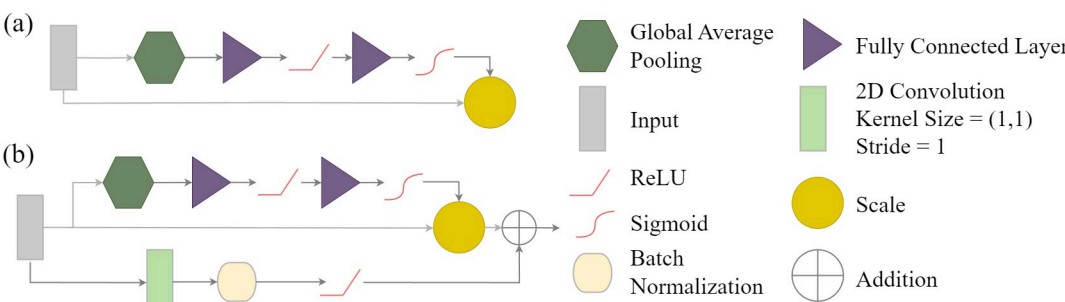

**Fig 7. Squeeze and excitation (SE) module after Hu et al. [71] optionally implemented between encoder blocks of geodl's UNet architecture.**

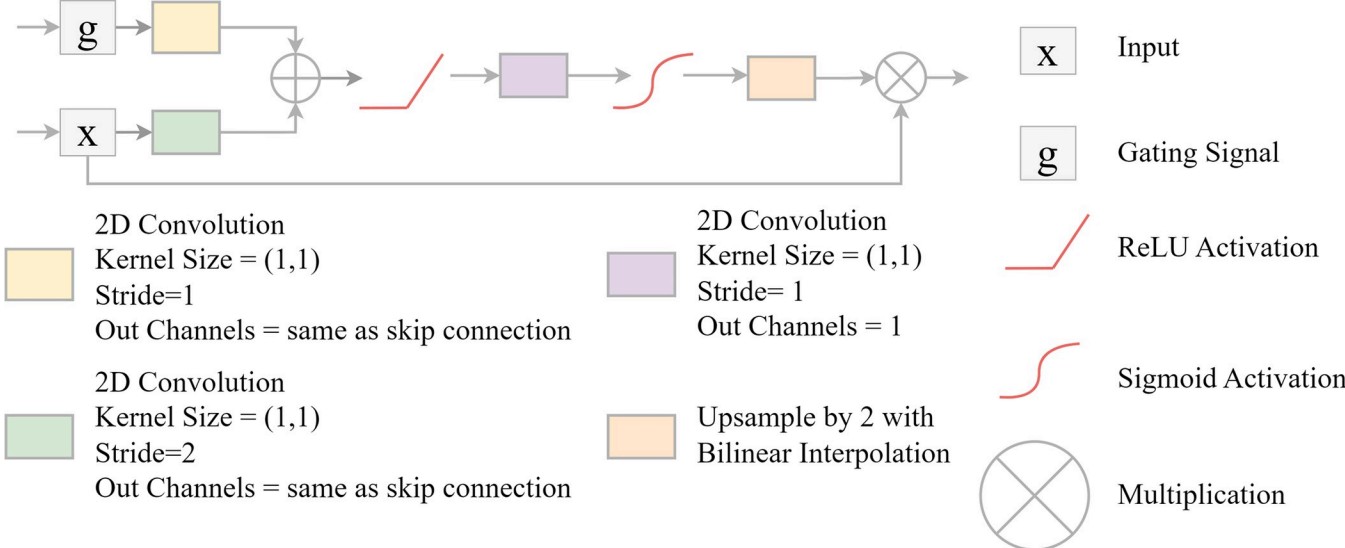

**Fig 8. Attention gate (AG) module after Abraham and Khan [72] and Oktay et al. [73] optionally implemented along skip connections of geodl's UNet architecture.**

that the size is the same as those from the next block. Batch normalization is then applied. The gating signal and augmented feature maps are then added together and passed through a ReLU activation. This result is then passed through a 1×1 2D convolution layer to produce a single output feature map. This feature map is then passed through a batch normalization layer followed by a sigmoid activation. In order to return the original spatial resolution of the input feature maps, upsampling is then applied using bilinear interpolation. The original feature maps from the encoder block of interest are multiplied by this upscaled result. Lastly, the results are concatenated with the upsampled feature maps from the next block, which are first upscaled using 2D transpose convolution, to be fed into the associated decoder block as normal [72, 73].

**Atrous spatial pyramid pooling (ASPP) module.**  The goal of atrous spatial pyramid pooling (ASPP) is to capture spatial context information at varying scales by increasing the size of the receptive field using dilated convolution. This technique is applied within the DeepLabv3+ architecture [74–76]. We implement a modified version of this module (Fig 9) as an optional replacement for the traditional UNet bottleneck block. It consists of performing dilated convolution using varying dilation rates. The results are then concatenated and passed through a 1×1 2D convolution layer to augment the number of total feature maps returned.

**Deep supervision.**  The goal of deep supervision is to offer additional training guidance by calculating auxiliary losses generated using predictions derived from feature maps generated in earlier stages of the architecture [77–80]. The feature maps produced by decoder blocks one, two, and three are upsampled to match the original resolution of the input data using bilinear interpolation. Next, 1×1 2D convolution is used to predict logits for each class at each pixel location using the feature maps generated by each decoder block separately. See Fig 5 above. These ancillary predictions are then used to calculate additional losses. Further, the user can control the relative weight of each of the four losses in the final loss calculation.

**UNet with MobileNet-v2 encoder.**  A second UNet model has also been included as part of this package. The *defineMobileUNet()* function defines a UNet architecture with a Mobile-Net-v2 backbone or encoder (Fig 10) [60, 81]. The MobileNet-v2 architecture is a lightweight CNN for use on mobile devices that incorporates many design innovations including depth-

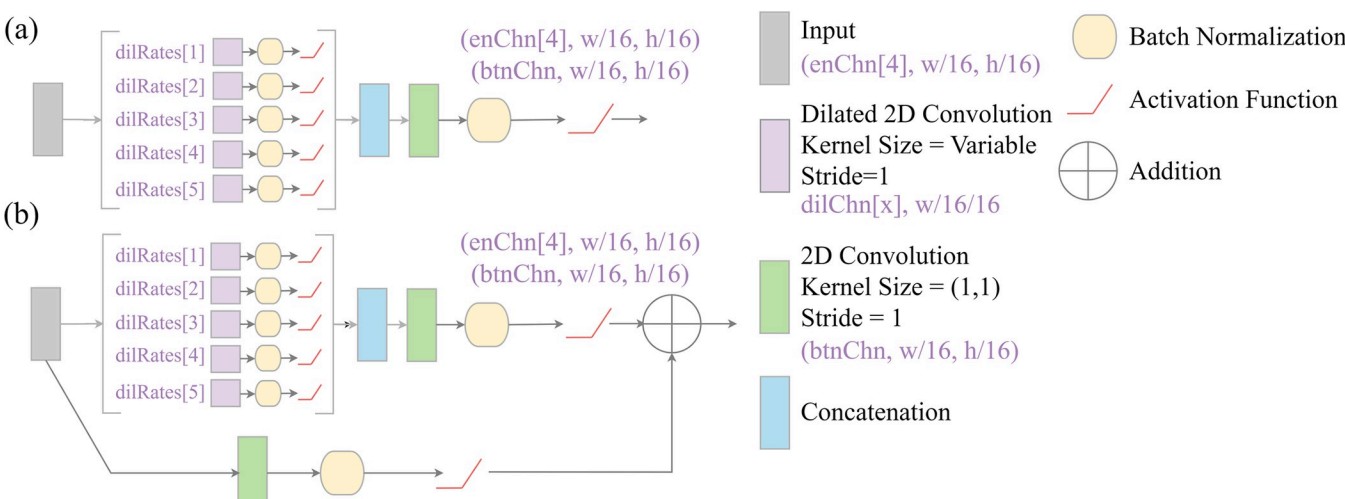

**Fig 9. Modified atrous spatial pyramid pooling (ASPP) module inspired by the DeepLabv3+ architecture and optionally implemented as a replacement bottleneck layer of geodl's UNet architecture.** w = width, h = height. See Table 3 for explanations of function parameter names. (a) conceptualizes an implementation without a residual connection while (b) includes a residual connection.

wise separable convolution and inverted residual and linear bottleneck layers [60]. This UNet implementation was inspired by a blog post by Sigrid Keydana [82, 83]. The model can be initialized using pre-trained weights based on ImageNet [84], and the encoder can be frozen (i.e., made not trainable during the learning process). Since this architecture makes use of ImageNet-based weights, and in contrast to our more general UNet implementation described above, it can currently only accept predictor variables with three input channels.

## Training, validation, and inference

**Loss metrics.** Cross entropy (CE) loss is generally the default multiclass classification loss metric [8]; however, alternative loss metrics have been proposed that can be especially useful when class proportions in the training set are imbalanced, which is a common occurrence in spatial predictive modeling, and/or when the user desires more control over the relative weightings of false positive (FP) and false negative (FN) errors relative to specific classes. The Dice (Eq 4) [85, 86] or Tversky (Eq 5) [87] loss is often used, which are generally termed region-based losses. The Tversky loss allows for specifying the relative weights of FN and FP errors using α and β terms, respectively. Dice- and Tversky-based losses make use of the rescaled class logits, obtained by applying a sigmoid or softmax activation, as opposed to the "hard" classification, as is the case when Dice, or the equivalent F1-score, is used as an accuracy assessment metric. Other options include macro-averaging, which gives equal weight in the aggregated assessment metric, or weighted macro-averaging, which allows the user to control the relative weight of the classes [85–89]. We do not implement micro-averaging, because that method is equivalent to overall accuracy, and thus is sensitive to class proportions [88].

Focal losses, such as focal CE, focal Dice, and focal Tversky, allow for adding additional weight to difficult-to-predict samples or classes, which are defined as those that have a low predicted rescaled logit for their correct class [72, 89, 90].

Multiclass macro−averaged Dice loss

$$= \frac{1}{N} \sum_{j=1}^{C} (1 - (\frac{(2 \times \Sigma \hat{p}_{TP}) + \varepsilon}{(2 \times \Sigma \hat{p}_{TP}) + \Sigma \hat{p}_{FN} + \Sigma \hat{p}_{FP} + \varepsilon})) \qquad (Eq4)$$

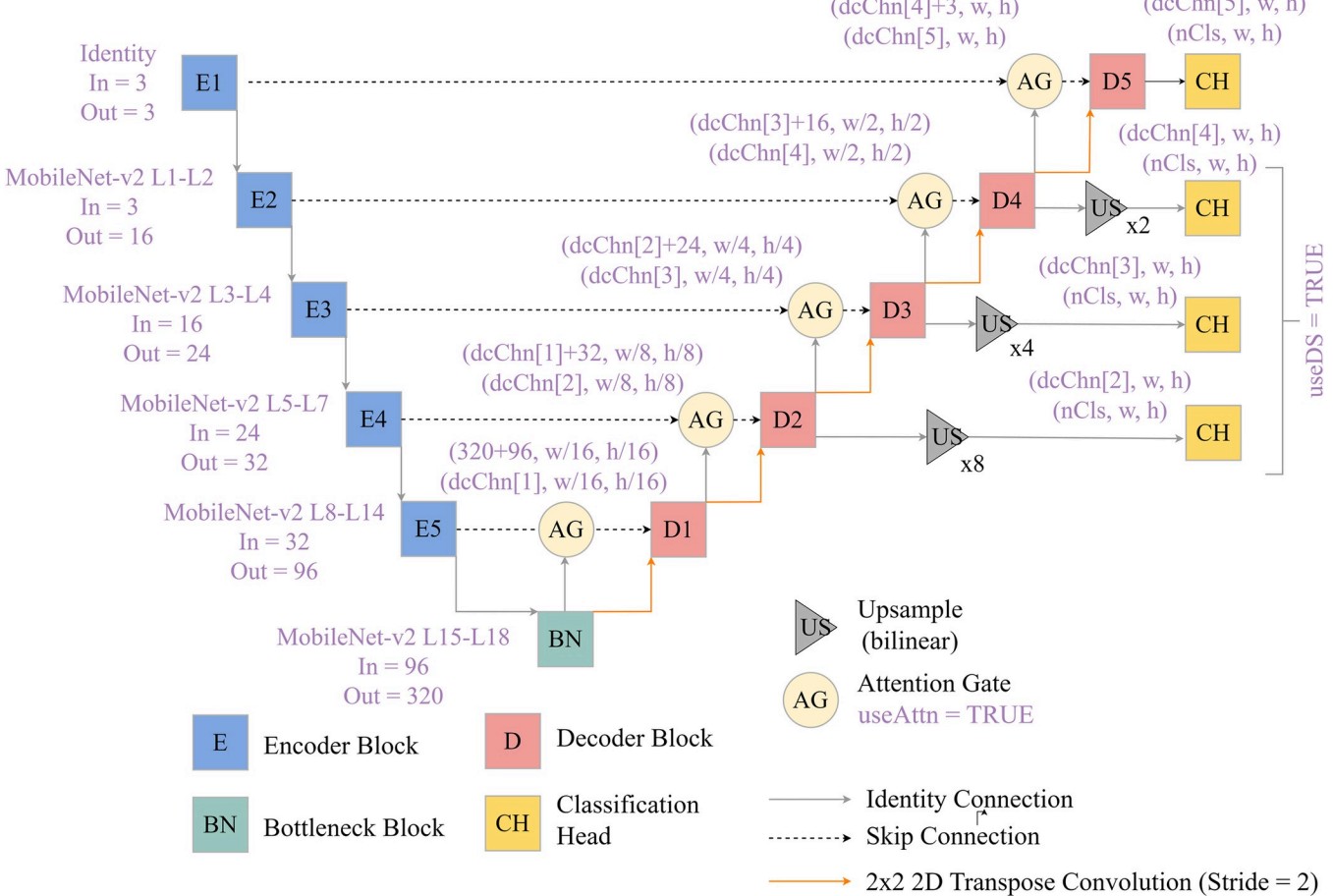

**Fig 10. UNet with MobileNet-v2 encoder, attention gates along skip connections, and deep supervision as implemented in geodl.** Blocks prefixed with E represent encoder blocks while those prefixed with D represent decoder blocks. Encoder blocks are derived from the MobileNet-v2 architecture. L = layer, w = width, h = height. See Table 3 for explanations of function parameter names.

$$\text{Multiclass Tversky loss} = \frac{1}{N}\sum\nolimits_{j=1}^{C}\left(1 - \left(\frac{\Sigma\hat{p}_{TP} + \varepsilon}{\Sigma\hat{p}_{TP} + \alpha\Sigma\hat{p}_{FN} + \beta\Sigma\hat{p}_{FP} + \varepsilon}\right)\right) \qquad \text{(Eq5)}$$

In order to expand the range of loss functions available, we implemented a modified version of the unified focal loss proposed by Yeung et al. [91]: *defineUnifiedFocalLoss()*. Modifications from the original implementation include: (1) allowing users to define separate class weights for both the distribution-based and region-based metrics, (2) using class weights as opposed to the symmetric and asymmetric methods implemented by the authors, and (3) including an option to apply a logcosh transform for the region-based loss, which can help stabilize the learning process by providing smoother gradients [92]. Eq 6 describes the implemented modified unified focal loss while the modified focal CE loss component is provided in Eq 7 and the modified Tversky loss is provided in Eq 8. The equation for the modified Tversky index, on which the modified Tversky loss is based, is provided in Eq 9.

$$\text{Modified unified focal loss} = \lambda \times \text{mFL} + (1 - \lambda) \times \text{mTL} \qquad \text{(Eq6)}$$

$$\text{Modified focal CE loss(mFL)} = -\frac{1}{\sum_{i=1}^{n} w_j} \sum_{i=1}^{n} \sum_{j=1}^{C} w_j \left[ (1 - \hat{y}_{ij})^{1-\gamma} \cdot y_{ij} \cdot \log(\hat{y}_{ij}) \right] \quad \text{(Eq7)}$$

$$\text{Modified Tversky loss(mTL)} = \frac{1}{\sum_{j=1}^{C} w_j} \sum_{j=1}^{C} w_j (1 - mTI_j)^{\gamma} \quad \text{(Eq8)}$$

$$\text{Modified Tversky Index(mTI)} = \left( \frac{\Sigma \hat{p}_{TP} + \varepsilon}{\Sigma \hat{p}_{TP} + \delta \Sigma \hat{p}_{FN} + (1 - \delta) \Sigma \hat{p}_{FP} + \varepsilon} \right) \quad \text{(Eq9)}$$

As described in Table 4, by adjusting the lambda ($\lambda$), gamma ($\gamma$), delta ($\delta$), and class weight terms (*clsWghtsDist* and *clsWghtsReg*), the user can implement a variety of different loss metrics. $\lambda$ controls the relative weight of the distribution- and region-based losses. If $\lambda = 0.5$, equal weighting between the losses is applied. If $\lambda = 1$, only the distribution-based loss is considered. If $\lambda = 0$, only the region-based loss is considered. $\gamma$ controls the application of focal loss and the application of increased weight to difficult-to-predict pixels (for the distribution-based loss) or difficult-to-predict classes (for the region-based loss). Smaller $\gamma$ values put increased weight on difficult samples or classes. Using a $\gamma$ of 1 equates to not using a focal adjustment. The $\delta$ term controls the relative weight of FP and FN errors for each class. The default is 0.6 for each class, which results in placing a higher weight on FN as opposed to FP errors [91].

**Assessment metrics.** The luz package provides the *luz_metric()* function to allow users to define new or custom metrics for use within training and validation loops [58]. The geodl package makes use of this function to create new implementations of recall, precision, and F1-score, which are not already implemented within luz. The geodl package also includes a version of the overall accuracy (OA) metric (*luz_metric_overall_accuracy()*) that accepts predictions and targets defined with the shapes and data types used within the package for standardization. Table 5 provides descriptions of the implemented assessment metrics. Macro-averaging is used in which the metric is calculated separately for each class and then averaged. Each class has equal weight in the resulting metric by default; however, users can choose to apply relative weightings. This is especially useful for binary classification problems when the user wishes to calculate precision, recall, and F1-score for only the positive case as opposed to averaging these metrics for both the positive and negative cases. As noted in Table 5 and in

**Table 4. Modified unified focal loss framework parameterization after Yeung et al. [91].** This framework is implemented by the *defineUnifiedFocalLoss()* function in geodl.

|  | Distribution-Based | Compound | Region-Based |
|---|---|---|---|
|  | $\lambda = 1$ | $\lambda < 1$ & $\lambda > 0$ | $\lambda = 0$ |
| $\gamma > 0$ & $\gamma < 1$ | Focal CE Loss | Unified Focal Loss | Focal Tversky Loss |
| $\delta \neq 0.5$ |  |  |  |
| $\gamma = 1$ | CE Loss | Tversky + CE Loss | Tversky Loss |
| $\delta \neq 0.5$ |  |  |  |
| $\gamma = 1$ | CE Loss | CE + Dice Loss | Dice Loss |
| $\delta = 0.5$ |  |  |  |

*clsWghtsDist* = relative weighting of classes in distribution-based loss (applied to each sample)

*clsWghtsReg* = relative weighting of classes in region-based loss (applied to each class when calculating a macro average)

*useLogCosH* = whether or not to apply a log cosh transformation to the region-based loss

**Table 5. Accuracy assessment metrics implemented by geodl using the *luz_mteric()* function from the luz package.**

| Metric | Function | Equation | Notes |
|---|---|---|---|
| Overall Accuracy (OA) | *luz_metric_overall_accuracy()* | $\frac{\text{Total Correct}}{\text{Total Samples}}$ | |
| Precision | *luz_metric_precision()* | $\frac{1}{N}\sum_{j=1}^{C}\frac{\text{TP}_j}{\text{TP}_j+\text{FP}_j}$ | Equivalent to average of class-level producer's accuracies |
| Recall | *luz_metric_recall()* | $\frac{1}{N}\sum_{j=1}^{C}\frac{\text{TP}_j}{\text{TP}_j+\text{FN}_j}$ | Equivalent to class-level user's accuracies |
| F1-Score | *luz_metric_f1score()* | $\frac{2\times\text{Recall}\times\text{Precision}}{\text{Recall}+\text{Precision}}$ | Harmonic mean of precision and recall |

alignment with terminology used in remote sensing, class-level recalls are equivalent to producer's accuracies (1 –omission error) while class-level precisions are equivalent to user's accuracies [93, 94].

**Other training and validation considerations.** We recommend using the luz package both to train and assess models, as implementing custom training and validation loops is error prone. The torch and luz documentation provides examples of training processes: https://torch.mlverse.org/. The geodl package webpage provides additional examples. The torch package provides access to many common optimization algorithms including mini-batch stochastic gradient descent [8, 57, 95, 96], Adagrad [97], Adadelta [98], RMSprop [99], Adam [100], and AdamW [101].

Generally, the learning rate is an important hyperparameter. One means to select a learning rate is described by Smith [102, 103]. The luz package provides an implementation of this learning rate finder method via the *lr_finder()* function [58]. luz also provides the *luz_callback_lr_scheduler()* function for defining and implementing callbacks to change or adapt the learning rate during the training process. The luz package provides additional callbacks that can be very useful during the learning process. For example, *luz_callback_early_stopping()* can be used to stop the learning process early if the model is no longer improving based on the loss or an assessment metric of interest. *luz_callback_csv_logger()* allows for logging calculated losses and metrics data to disk as a CSV file. *luz_callback_model_checkpoint()* can be used to save models to disk after each epoch or only if the model has improved based on the loss or an assessment metric [58].

**Model assessment and spatial predictions.** The *viewBatchPreds()* function allows for visualizing a mini-batch of predictions, reference masks, and predictor variables that were created using *defineSegDataSet()* and a data loader, and subsequently predicted with a trained model. geodl provides an *assessDL()* function for calculating assessment metrics from a dataloader. It also provides the *assessPnts()* function, which allows for performing assessments at point locations, and the *assessRaster()* function, which allows for assessment using entire raster grids.

These functions generate a set of summary metrics when provided reference and predicted classes. Alongside the complete confusion matrix, the following metrics are calculated: OA, average class user's accuracy (i.e., precision), average class producer's accuracy (i.e., recall), and average class F1-score. For average class user's accuracy, producer's accuracy, and F1-score, macro-averaging is used where all classes are equally weighted [88, 104, 105]. All class user's and producer's accuracies are also returned. For assessing map output, we generally recommend using a testing set that honors the true landscape proportions of each class. When a confusion matrix is generated using proportions that approximate the true landscape proportions, it is termed an estimated population confusion matrix [106].

The *predictSpatial()* function allows for predicting to a raster extent. In order to process large raster extents, chips are extracted relative to the *chpSize* parameter. Overlap between chips is specified using the *stride_x* and *stride_y* parameters. We generally recommend using an overlap of at least 25% between adjacent chips. It has generally been found that predictions nearer to the margin of a chip have lower accuracy than those in the interior of the chip, likely due to the lack of a full set of neighboring pixels. As a result, the *crop* parameter can be set to remove outer rows and columns of pixels and not include them in the final, merged product. Using an overlap via the *stride_x* and *stride_y* parameters in combination with cropping (*crop*) allows for only predictions in the center of each processed chip to be included in the final, merged product. The *predType* parameter controls the type of prediction returned, either class numeric codes (i.e., "hard" labels), class logits, or logits rescaled using a softmax function. It is also important that data being predicted be rescaled and/or normalized using the same settings defined for the training dataset.

## Case studies

In this section, we present some sample classification results obtained using geodl. All experiments were conducted on a Windows-based workstation with an Intel i7 2.5 GHz processor, 64 GB of RAM, and a single GeForce RTX 3060 GPU with 12GB of VRAM. Two different experiments were conducted. First, models were trained for 25 epochs using a larger training set: 3,886 samples for the topoDL datasets and 4,000 samples for the landcover.ai dataset. Second, model replicates were tested using different random 2,000 sample training subsets for the topoDL problem and random 3,000 sample training subsets for the landcover.ai problem. A smaller set was used to reduce computational time and to document model variability when different random subsets of the data are used and models are initialized using different random seeds. For all experiments, the epoch that provided the lowest loss for the validation set was selected as the final model and used to predict to the withheld test data to calculate assessment metrics. When using the larger training sets, a single training and associated validation epoch took between 10 and 20 minutes to run for the topoDL problem and 1 and 1.5 hours to execute for the landcover.ai problem. We attribute the longer runtime for the lancover.ai models to the larger chip size, 512-by-512 cells as opposed to 256-by-256 cells, and the more complex model architecture that integrated the MobileNet-v2 backbone.

### Surface mine disturbance from historic topographic maps

Within our larger training set example for the topoDL problem, 3,886 256-by-256 cell image chips were used to train the model while 812 chips were used to validate the model at the end of each training epoch and 1,246 were maintained as a test set to assess the final model. The data were partitioned such that all chips from the same topographic map were included in the same data partition in order to avoid spatial autocorrelation between the data partitions. For the training set, a maximum of two augmentations were performed per chip, vertical or horizontal flips, with a 0.75 probability of being applied. In other words, up to two augmentations were applied to a single chip, and there was a 75% chance of applying each augmentation. If no augmentation is applied, the original chip is returned. A mini-batch size of 15 was used, and the model was trained for 25 epochs. The model that returned the lowest loss for the validation data was maintained as the final model, which was the model state after epoch 20. The modified unified focal loss was configured as a Tversky loss using $\lambda = 0$, $\gamma = 1$, and $\delta = 0.6$. In other words, only the region-based loss was included with no focal adjustment but unequal weighting for FN and FP errors per class. The AdamW optimizer was used with a learning rate of 1e-3, which was selected using the learning rate finder implementation in the luz package.

**Table 6. Confusion matrix and derived metrics for topoDL [52] classification.**

| | | Reference | | |
|---|---|---|---|---|
| | | **Background** | **Mine** | |
| **Prediction** | Background | 69,762,584 | 326,278 | NPV = 0.995 |
| | Mine | 527,709 | 10,975,749 | Precision = 0.954 |
| | | Specificity = 0.993 | Recall = 0.971 | F1-Score = 0.963 |
| | | | | OA = 0.990 |

Values represent counts of pixels or cells. OA = overall accuracy; NPV = negative predictive value.

Table 6 provides the confusion matrix and derived metrics calculated for the withheld test set and using the *assessDL()* function while Fig 11 provides an example set of image chips (a), reference masks (b), and predictions (c) for a mini-batch of test samples created using the *viewBatchPreds()* function. The OA for the prediction was 0.990 while the F1-score was 0.963. The precision was 0.954 while the recall was 0.971, suggesting a good compromise between commission and omission errors. Table 7 shows summary statistics for the assessment metrics calculated using ten model replicates, different random seeds, and different random subsets of 2,000 training chips. Again, this second experiment was conducted to document performance variability and is not meant to indicate best performance since a subset of the training chips were used and each model replicate was trained for up to only ten epochs.

## Landcover.ai multiclass land cover classification

For the landcover.ai experiment, 4,000 512-by-512 cell image chips were used to train the model while 1,000 chips were used to validate the model at the end of each training epoch and

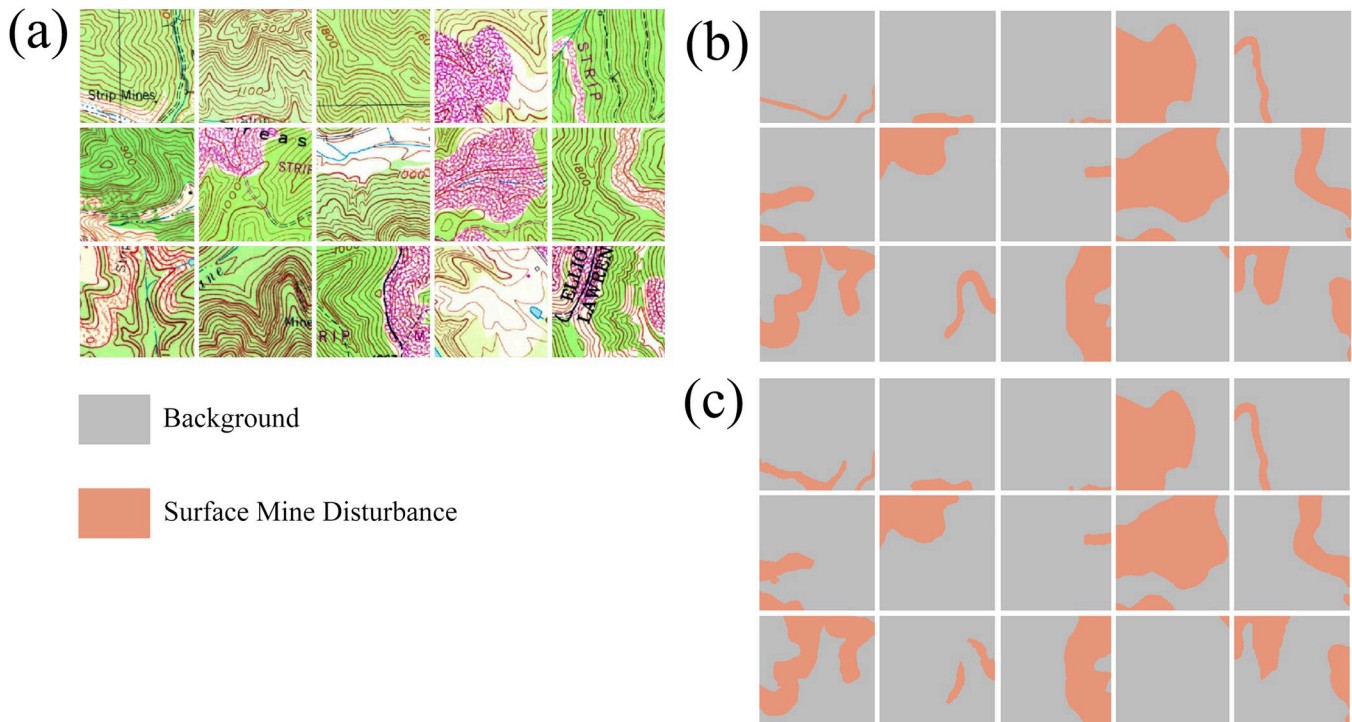

**Fig 11. Example prediction of surface mine disturbance extents from topoDL dataset [52].** (a) example image chips; (b) reference labels or masks, (c) predictions generated using geodl workflow.

**Table 7. Model assessment metrics based on ten model replicates with different random seeds and training subsets.**

|         | OA    | F1-score | Recall | Precision | Specificity | NPV   |
|---------|-------|----------|--------|-----------|-------------|-------|
| Mean    | 0.981 | 0.930    | 0.938  | 0.923     | 0.987       | 0.990 |
| Median  | 0.983 | 0.937    | 0.941  | 0.935     | 0.990       | 0.991 |
| SD      | 0.004 | 0.014    | 0.023  | 0.023     | 0.004       | 0.004 |
| Minimum | 0.972 | 0.903    | 0.892  | 0.864     | 0.976       | 0.983 |
| Maximum | 0.985 | 0.947    | 0.968  | 0.940     | 0.991       | 0.995 |

OA = overall accuracy; NPV = negative predictive value.

792 were maintained as a testing set to assess the final model. Data partitions were defined by the data originators. For the training set, a maximum of two augmentations were performed per chip, vertical or horizontal flips, with a probability of 0.75 of being applied. A mini-batch size of 20 was used, and each model was trained for 25 epochs. The model that returned the lowest loss for the validation data, the model state after epoch 21, was maintained as the final model. The modified unified focal loss was configured as a focal Dice loss using $\lambda = 0$, $\gamma = 0.8$, and $\delta = 0.5$. In other words, only the region-based loss was included, a focal adjustment was applied to increase the relative cost of misclassifying difficult classes, and FN and FP errors per class were equally weighted. The AdamW optimizer was used with a learning rate of 1e-3, which was selected using the learning rate finder implementation in the luz package.

Table 8 provides a confusion matrix, class-level user's and producer's accuracies, and class-level F1-scores calculated for the withheld test set and using the *assessDL()* function while Fig 12 provides an example set of image chips (a), reference masks (b), and predictions (c) for a mini-batch of testing samples created using the *viewBatchPreds()* function. The OA was 0.908 and the macro-averaged, class-aggregated F1-score was 0.823. The woodland class showed both the lowest producer's and user's accuracies. Table 9 provides the OA and macro-averaged, class-aggregated F1-score (aF1), producer's accuracy (recall) (aPA), and user's accuracy (precision) (aUA) summary metrics calculated for the ten model replicates using different random seeds and 3,000-sample subsets of the available training chips. Again, this second experiment was conducted to document variability and is not meant to indicate best performance since a subset of the training chips were used and each model replicate was trained for only ten epochs.

## Conclusions and future development

The goal of geodl is to provide a complete workflow as an R-based tool to perform DL-based semantic segmentation that adheres to standards and best practices within geospatial

**Table 8. Confusion matrix and class-level user's and producer's accuracies for landcover.ai [53] classification.** Overall accuracy = 0.908, macro-averaged producer's accuracy = 0.885, macro-averaged user's accuracy = 0.770, and macro-averaged F1-score = 0.823.

|            |                    | Reference   |          |          |           |            |                 |
|------------|--------------------|-------------|----------|----------|-----------|------------|-----------------|
|            |                    | Background  | Building | Woodland | Water     | Road       | User's Accuracy |
| Prediction | Background         | 102,511,856 | 197,898  | 381,378  | 561,041   | 2,120,744  | 0.969           |
|            | Building           | 588,494     | 1,843,051| 23,588   | 3,089     | 5,063      | 0.748           |
|            | Woodland           | 4,356,259   | 46,980   | 2,984,637| 15,177    | 153,713    | 0.395           |
|            | Water              | 1,386,604   | 13,854   | 27,477   | 9,723,005 | 295,632    | 0.849           |
|            | Road               | 8,400,555   | 27,141   | 402,337  | 86,647    | 71,461,828 | 0.889           |
|            | Producer's Accuracy| 0.874       | 0.866    | 0.781    | 0.936     | 0.965      |                 |
|            | F1-score           | 0.919       | 0.803    | 0.525    | 0.891     | 0.926      |                 |

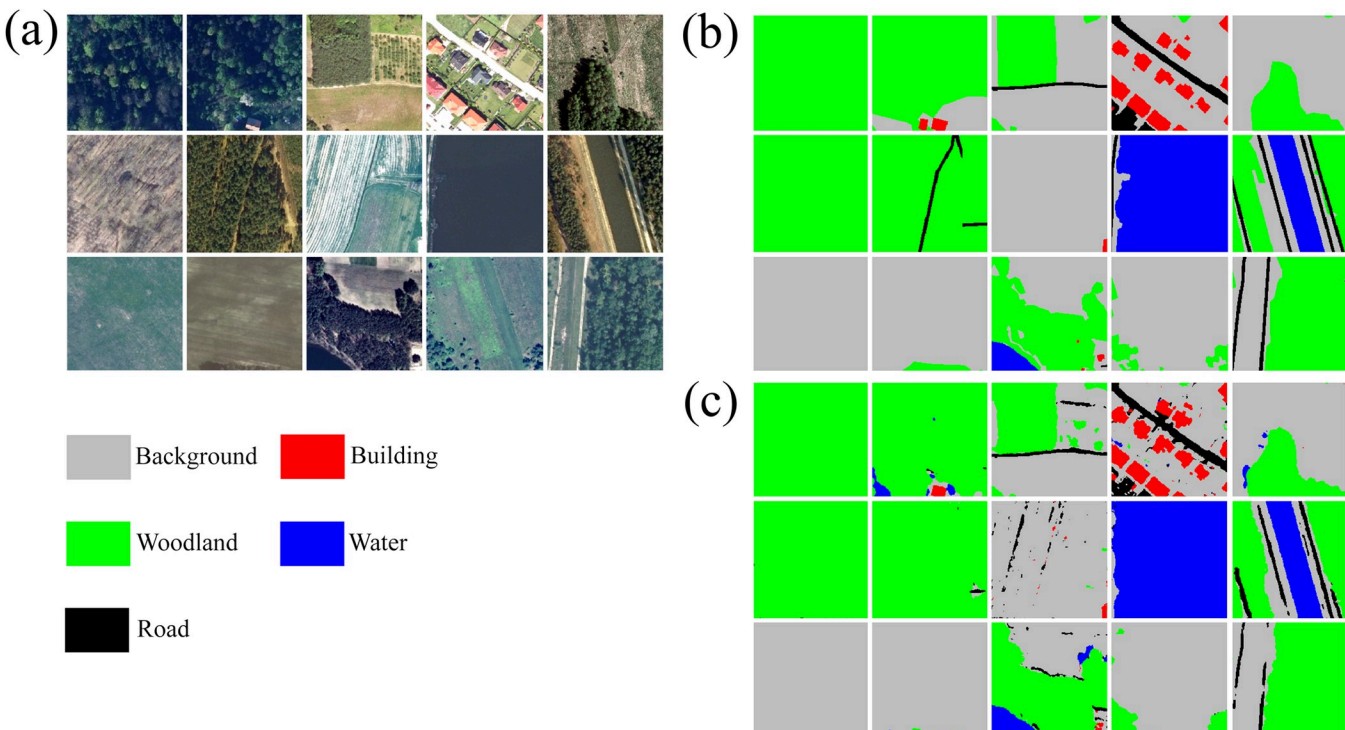

**Fig 12. Example prediction of general land cover from landcover.ai dataset [53].** (a) example image chips; (b) reference labels or masks, (c) predictions generated using geodl workflow.

predictive modeling and remote sensing. The use of the torch package simplifies the software stack since it is not necessary to interface with a Python environment and associated libraries. It also supports the use of GPU-based computation, which is necessary for practical use of DL applied to large datasets. The use of terra allows for efficient handling of large raster datasets with varying number of bands. Lastly, luz greatly simplifies the DL training and validation processes and the placement and transfer of models and data between the CPU and GPU. We argue that geodl provides an intuitive workflow applicable to a wide variety of geospatial semantic segmentation problems and input data that can be represented as multidimensional arrays. It makes well established DL workflows and UNet-like architectures available to geospatial, remote sensing, and Earth scientists, and is particularly useful for users who are more comfortable in the R environment than other languages, such as Python.

**Table 9. Overall accuracy and macro-averaged class aggregated assessment metrics for landcover.ai [53] classification using ten replicates and different 3,000 chip random data partitions.**

|  | OA | aF1 | aPA | aUA |
|---|---|---|---|---|
| Mean | 0.921 | 0.829 | 0.816 | 0.843 |
| Median | 0.922 | 0.831 | 0.824 | 0.847 |
| SD | 0.004 | 0.020 | 0.021 | 0.025 |
| Minimum | 0.915 | 0.774 | 0.763 | 0.786 |
| Maximum | 0.928 | 0.852 | 0.836 | 0.881 |

OA = overall accuracy, aF1 = macro-averaged, class aggregated F1-score, aPA = macro-averaged, class aggregated producer's accuracy (recall), aUA = macro-averaged, class aggregated user's accuracy (precision).

We plan to further develop geodl with future releases. First, we plan to implement a torch dataset subclass that allows for sampling from larger raster grids dynamically as opposed to generating image chips, similar to the implementation in the TorchGeo Python package [25]. We plan to implement additional CNN-based models including UNet3+ [80] and DeepLabv3 + [74, 75, 107]. Generally, we would like to provide a wider range of semantic segmentation algorithms and backbones, similar to the Segmentation Models Python package [23, 24]. Additional development of loss functions would also be valuable, such as the ability to weight pixels based on their distance from class boundaries or select subsets of pixels for loss calculations. We would also like to expand the package to include transformer-based segmentation DL architectures, such as SegFormer [108]. We plan to develop additional functions for customizing the training loop, such as using different learning rates for different components of the model architecture. We intend for the package to maintain its focus on geospatial semantic segmentation and do not plan to implement scene classification, object detection, and instance segmentation methods. We are interested in finding others to contribute to the package. Ultimately, we hope that geodl is a useful contribution to the torch ecosystem in R.

## Author Contributions

**Conceptualization:** Aaron E. Maxwell, Srinjoy Das.

**Data curation:** Aaron E. Maxwell.

**Formal analysis:** Aaron E. Maxwell.

**Funding acquisition:** Aaron E. Maxwell.

**Investigation:** Aaron E. Maxwell, Sarah Farhadpour, Srinjoy Das.

**Methodology:** Aaron E. Maxwell.

**Project administration:** Aaron E. Maxwell.

**Software:** Aaron E. Maxwell, Sarah Farhadpour, Yalin Yang.

**Supervision:** Aaron E. Maxwell.

**Writing – original draft:** Aaron E. Maxwell.

**Writing – review & editing:** Aaron E. Maxwell, Sarah Farhadpour, Srinjoy Das.

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
