## [Decision Letter · Decision Letter 0]

4 Oct 2024

PONE-D-24-36665geodl: An R package for geospatial deep learning semantic segmentation using torch and terraPLOS ONE

Dear Dr. Maxwell,

Thank you for submitting your manuscript to PLOS ONE. After careful consideration, we feel that it has merit but does not fully meet PLOS ONE’s publication criteria as it currently stands. Most of reviewers gave positive comments. Therefore, we invite you to submit a revised version of the manuscript that addresses the points raised during the review process.

We look forward to receiving your revised manuscript.

Kind regards,

Xiaoyong Sun

Academic Editor

PLOS ONE

sunx1@sdau.edu.cn

Journal Requirements:

Reviewers' comments:

Reviewer's Responses to Questions

**Comments to the Author**

1. Is the manuscript technically sound, and do the data support the conclusions?

Reviewer #1: Partly

Reviewer #2: Yes

Reviewer #3: Yes

Reviewer #4: Yes

2. Has the statistical analysis been performed appropriately and rigorously? 

Reviewer #1: Yes

Reviewer #2: N/A

Reviewer #3: Yes

Reviewer #4: Yes

3. Have the authors made all data underlying the findings in their manuscript fully available?

Reviewer #1: Yes

Reviewer #2: Yes

Reviewer #3: Yes

Reviewer #4: Yes

4. Is the manuscript presented in an intelligible fashion and written in standard English?

Reviewer #1: Yes

Reviewer #2: Yes

Reviewer #3: Yes

Reviewer #4: Yes

5. Review Comments to the Author

Reviewer #1: Overall comments

First of all, I am a big fan of software papers, and think they need more emphasis in academia as some of the most valuable contributions researchers can make. I'm glad to see the package now available on CRAN (already with ~350 downloads since its release), resolving most of the concerns of prior reviewers.

I am coming from a remote sensing and deep learning background, although from the Python/PyTorch side of things. I have little to no experience with R or even with PLOS ONE, so forgive me if anything I say is out of touch.

I really enjoyed reading this paper, especially pages 1–18 and 35–39. However, I believe pages 19–34 could be deleted without negatively impacting the quality of the paper.

When I started reading this paper, I was really excited about learning how geodl was implemented and how it differs from other related works. However, there is very little discussion of related works other than TorchGeo. In the Python space, Raster Vision and eo-learn are also worth mentioning, while in R, the only library I could find is SITS, which sacrifices spatial support for time-series support but still involves geospatial ML.

Pages 19–34 are essentially a deep learning textbook. There are little to no novel contributions introduced here, it is simply reference material for someone looking to learn more about deep learning. I don't know what PLOS ONE papers usually look like, but I'm used to short ML conference papers that rarely ever approach 50% background material. Like the previous reviewers, it is unclear to me what the audience for this paper is. People unfamiliar with deep learning will likely need to take a machine learning course before using your model, and people familiar with deep learning already know everything on pages 19–34.

Comments on software

In all honesty, the paper itself is more of a formality to me; the software is really what's important. In general, I don't care that much about the quality of the paper, but I want to make sure the software is well maintained and has promise.

Although the paper goes into extensive detail about the design of geodl, there is little information about its capabilities. For example, which datasets have been implemented? Which encoder and decoder model architectures? How many pre-trained model weights are available? Are all model weights pre-trained on ImageNet, or are there any actually pre-trained on remote sensing data? Is time-series data supported? Are there any plans to support classification, regression, pixelwise regression, object detection, instance segmentation, change detection, or any other common tasks in computer vision and remote sensing?

I'm not familiar with R, so perhaps this is a naive question, but where is your CI/CD? Where can I find your unit tests? What percentage of the code is covered by tests? I can't, in good faith, recommend that anyone use software with less than 80% unit test coverage. For comparison, Raster Vision, eo-learn, and SITS all have over 90% unit test coverage, while TorchGeo has 100% coverage.

For the assessment metrics, it is unclear if these are part of geodl or luz. If they are not part of geodl, they shouldn't be in this paper in my opinion.

Comments on experiments

Datasets: Geodl has a number of exciting applications for remote sensing and Earth observation practitioners. However, experiments are only shown for RGB datasets, despite the vast majority of remote sensing data being multispectral, hyperspectral, thermal, or radar (SAR, inSAR). I would much rather see experiments on these data modalities, as they are more realistic for what users will actually need. Are these modalities even supported by the models you implement, or are they RGB-only?

Augmentations: The experimental design is brief but riddled with mistakes. The authors choose a maximum of 1 augmentation per image, and images are only augmented 50% of the time. However, this is not how computer vision (or deep learning in general) are done. Augmentations like random horizontal and vertical flip should each be applied 50% of the time and in sequence, such that 25% are the normal orientation, 25% are horizontally flipped, 25% are vertically flipped, and 25% are both horizontally and vertically flipped. This results in the maximum possible dataset diversity, especially since overhead aerial images have no natural orientation direction. Similarly, random rotation should be applied 100% of the time, with orientations ranging from 0–360º. Modifications to things like saturation are not defined for multispectral imagery, and generally hurt performance on RGB satellite imagery where the spectral signature is important to distinguish similar classes. In the LandCover.ai experiments, the authors use: "either a vertical flip, horizontal flip, brightness adjustment, or saturation adjustment with probabilities of being applied of 0.5, 0.5, 0.1, and 0.2, respectively". However, these probabilities do not add to 1. It looks like prior reviewers also noticed this mistake, and it was fixed in the topoDL section but not the LandCover.ai section.

Compute: There is no mention of the computational resources used in this paper. What model of GPUs and how many GPUs were used? The model was trained for 10 epochs, but how long did that take (in hours)? This is important for comparison with other implementations to see if there are bottlenecks in the model code or in I/O.

Metrics: It's customary to provide uncertainty quantification in your reported metrics. In tables 6–8, this could be as simple as running your experiment multiple times with different PRNG seeds and reporting the mean and std dev of accuracy/precision/recall/F1. Without these, it's difficult to tell if the model always works reliably or if you just cherry-picked a good random seed.

Comments on writing

The overall writing quality is extremely high and practically devoid of typos and grammar mistakes. My only complaint is the percentage of the paper that feels like well-known background material that can be found in any deep learning textbook.

I would add more detailed captions to all figures and tables. Each figure and table should be standalone and self-explanatory, even without the surrounding text.

* Table 1: I would change the purpose of "validation data" from "assess model generalization" to "evaluate hyperparameter choices". Again, all of these definitions are standard. I don't think it's possible to pass a single ML course without knowing these.

* Figure 1/2: The diagram makes it look like geodl only supports vector masks. It may be good to clarify whether or not geodl also supports raster masks, and update the diagram to show multiple possible starting points if that is the case.

* Equation 3: This equation is not in parentheses, unlike all other equations.

Reviewer #2: Thank you for the impressive amount of effort you put into not just preparing your package to pass CRAN checks and requirements, but also in developing multiple vignettes, example datasets, and clear documentation.

From my perspective as an experienced R user with interest in but no experience with geospatial DL, I found the reorganized manuscript much easier to follow. Clearly separating the sections into general overview and more technical material, and adding more explanation (like Table 1), was very helpful for me. Figures 1 and 2 support this organization, with both high-level and more detailed workflows that clearly denote functions included in this package vs those from other important packages.

This is a challenging but very hot topic, and I think that your package will be of great interest.

Reviewer #3: The manuscript is a resubmission of the authors' previous work. In this revised version, the authors have improved the clarity of the manuscript, published the R package on CRAN, and satisfactorily addressed the issues raised by my previous review.

Reviewer #4: This paper introduces a R package for training deep learning semantic segmentation models with geospatial data and shows two applications of this package with existing benchmark datasets from the literature. This is a commendable and important effort that fills a large gap in functionality in the R ecosystem -- enabling researchers in scientific fields where R is a standard to work with modern ML approaches for semantic segmentation. Modern semantic segmentation models, along with the necessary training and testing pipelines to use them are non-trivial to implement correctly (further, when implemented incorrectly they can still seem to work for many reasons). Scientific users should be abstracted away from the details of implementation in order to focus on their specific problem and this cross-cutting infrastructure work is important to making this happen. The acceptance of the package on CRAN is a great milestone for this work!

From a results standpoint, the experiments on LandCoverAI show performance that is similar to other approaches reported in the literature. The experiments on the surface mines give a strong benchmark that other methods can attempt to beat and that represent a usable approach to detecting surface mines in practice.

My only minor comments would be to condense the writing where possible (the current manuscript is very long), consider adding space or larger borders between patches in Fig 3, 4, 11, 12, consider removing the architecture diagrams in favor of citing the original work (I'm not sure these improve understanding of the method or help make a point other than, "the models are non-trivial to implement"), and consider a careful proofreading (e.g. in the abstract I might edit, "Such methods are especially applicable to pixel-level classification, or semantic segmentation, tasks."  "Such methods are especially applicable to pixel-level classification or semantic segmentation tasks.").

6. PLOS authors have the option to publish the peer review history of their article (what does this mean?). If published, this will include your full peer review and any attached files.

Reviewer #1: No

Reviewer #2: No

Reviewer #3: No

Reviewer #4: No

---

## [Author Response · Author response to Decision Letter 0]

14 Oct 2024

We have uploaded a response to reviewers document.

---

## [Decision Letter · Decision Letter 1]

21 Nov 2024

geodl: An R package for geospatial deep learning semantic segmentation using torch and terra

PONE-D-24-36665R1

Dear Dr. Maxwell,

We’re pleased to inform you that your manuscript has been judged scientifically suitable for publication and will be formally accepted for publication once it meets all outstanding technical requirements.

Kind regards,

Xiaoyong Sun

Academic Editor

PLOS ONE

sunx1@sdau.edu.cn

Additional Editor Comments (optional):

Reviewers' comments:

Reviewer's Responses to Questions

**Comments to the Author**

1. If the authors have adequately addressed your comments raised in a previous round of review and you feel that this manuscript is now acceptable for publication, you may indicate that here to bypass the “Comments to the Author” section, enter your conflict of interest statement in the “Confidential to Editor” section, and submit your "Accept" recommendation.

Reviewer #1: All comments have been addressed

2. Is the manuscript technically sound, and do the data support the conclusions?

Reviewer #1: Yes

3. Has the statistical analysis been performed appropriately and rigorously? 

Reviewer #1: Yes

4. Have the authors made all data underlying the findings in their manuscript fully available?

Reviewer #1: Yes

5. Is the manuscript presented in an intelligible fashion and written in standard English?

Reviewer #1: Yes

6. Review Comments to the Author

Reviewer #1: Responses to author rebuttal:

> Unless the editor objects, we would prefer to maintain this section of the paper.

If you want to keep it, it's fine with me, I just don't think it's necessary. All research (both papers and software) builds on prior work. This is the purpose of citations and references, to point the reader in the right direction if they are unfamiliar with the necessary background. If you want to teach the reader everything they need to know about deep learning and semantic segmentation, that's fine, although it may make more sense as an appendix.

> We have edited the Needs and Justification section to mention these other packages/tools (i.e., Raster Vision, eo-learn, and SITS)

eo-learn is for the scikit-learn ecosystem, not the PyTorch ecosystem. It's good that you have cited related works, although it would be nice to see a more detailed comparison that might convince readers why they should use your library instead of one of these existing ones.

> we feel that this material presented in pages 19-34 are essential to documenting the package

If you have made modifications to the vanilla U-Net architecture, then I agree that these are worth documenting. I'm not sure if these modifications warrant 16 pages of the paper, but I will leave that up to the authors and editors to decide.

> We do not plan to implement any specific datasets into geodl.

Thanks for clarifying this. From what you've described, it sounds like geodl will be quite useful for the remote sensing community, where the primary focus is on working with raw, uncurated raster and vector data layers. However, without specific implementations of data loaders for curated benchmark datasets, I feel this library will fall short of what is needed by the broader machine learning community. That is perfectly fine of course, there is no reason you need to support both communities.

> The defineUNet() implementation is meant to be trained from scratch. The defineMobileUNet() implementation can be initialized using ImageNet weights; however, it currently only accepts three-band input data.

This is unfortunate. Three-band geospatial data is rather uncommon, and there are no satellite images in ImageNet. Although transfer learning starting from ImageNet pre-trained weights performs better than expected, numerous papers have shown that in-domain pre-training on satellite imagery significantly improves performance on downstream tasks in remote sensing. The vast majority of potential users of your software will not have enough experience with self-supervised learning to create their own pre-trained weights, nor will they have sufficient data to train models from random initializations. While I believe your library still has the potential for use with high resolution imagery (which is often RGB or RGB+NIR-only), I suspect that it will not be very useful to the majority of remote sensing researchers.

> The goal is for this package to focus on semantic segmentation. We do not plan to further expand it to other types of tasks, such as scene classification, object detection, instance segmentation, or regression.

This is fine, semantic segmentation is definitely the most common application for remote sensing. Although multispectral imagery like Landast or Sentinel-2 is more commonly used than high resolution imagery, and you primarily support RGB-only data.

> We plan to further develop geodl with future releases.

Thanks for providing this detailed Future Works section! Software papers are always starting points. You don't need to have everything complete before you publish your paper. The important thing is a well-defined scope, a clear gap in existing software libraries, and enough evidence of good software practices to ensure the longevity of the project.

> CRAN has strict requirements for getting a package formally accepted.

I'm aware that CRAN has much stricter requirements for publishing than PyPI. However, these requirements are primarily focused on packaging (ensuring that all dependencies are correctly listed) and documentation. What I'm worried about here is unit testing, which ensures that the software does what it is expected to do, on a variety of R versions, dependency versions, and operating systems. Any manual checks done by CRAN maintainers are limited by the time these maintainers spent analyzing every line of code (not possible for large packages) and by their own area of expertise (they may not know anything about deep learning). Continuous integration is necessary to ensure that the software is not only bug free, but that it remains bug free. Without this, it is merely research code, not software.

> We have re-executed both examples.

Thank you for re-running all experiments! Experiment 2 looks great for uncertainty quantification, and all standard deviations are low as expected.

> We also changed how augmentations were performed. Now, up to two random augmentations, each with a 0.75 chance of being applied, were implemented.

This is still incorrect. Each random flip should have a 50% chance of being applied (for overhead remote sensing data) in order to maximize dataset diversity.

> We have added information about our computational environment at the beginning of the Case Study section

Looks perfect, thanks!

> Since Equation 3 does not have a condition, we don’t think parentheses are necessary.

I'm not sure what conditionals have to do with parentheses. To clarify, I'm suggesting to change "Equation (3)" to "(Equation 3)" so as to match all other equations.

Overall comments:

I would like to thank the authors for responding to all of my unnecessarily long and detailed review comments. I know how much work was involved in rerunning all experiments and clarifying what the library does or does not support. It is now clear to me that geodl represents the state of the art in geospatial machine learning in the R ecosystem.

With that said, geodl is ~5–10 years behind what would be considered state of the art in Python/PyTorch. Until the following highly desirable features are implemented, I don't see very many people adopting this library by choice:

1. Data loaders for curated benchmark datasets: required for adoption by the machine learning community

2. Pre-trained models for multispectral, hyperspectral, and radar data: required for adoption by the remote sensing community

3. Continuous integration and unit testing: required for confidence in the library and growing the number of contributors

My general opinion is that this paper should be accepted. While I don't necessarily see the value of this library when looking at the overall deep learning landscape, if you put a gun to my head and forced me to use R, I would probably use this library.

I believe the usefulness of my reviews have plateaued. If this paper is accepted, I would encourage the authors to correct a few minor remaining issues:

* eo-learn is for scikit-learn, not PyTorch

* Equation (3) should be (Equation 3)

* Random flips should each be applied 50% of the time

7. PLOS authors have the option to publish the peer review history of their article (what does this mean?). If published, this will include your full peer review and any attached files.

Reviewer #1: No

---

## [Editor Report · Acceptance letter]

25 Nov 2024

PONE-D-24-36665R1 

PLOS ONE

Dear Dr. Maxwell, 

I'm pleased to inform you that your manuscript has been deemed suitable for publication in PLOS ONE. Congratulations! Your manuscript is now being handed over to our production team.

Kind regards, 

on behalf of

Dr. Xiaoyong Sun 

Academic Editor

PLOS ONE